# All-fiber high-speed image detection enabled by deep learning

Zhoutian Liu[1], Lele Wang [1], Yuan Meng [1], Tiantian He[1], Sifeng He[1], Yousi Yang[1], Liuyue Wang[1], Jiading Tian [1], Dan Li[1,2], Ping Yan [1,2], Mali Gong[1,2], Qiang Liu[1,2] & Qirong Xiao [1,2 ✉]

Ultra-high-speed imaging serves as a foundation for modern science. While in biomedicine, optical-fiber-based endoscopy is often required for in vivo applications, the combination of high speed with the fiber endoscopy, which is vital for exploring transient biomedical phenomena, still confronts some challenges. We propose all-fiber imaging at high speeds, which is achieved based on the transformation of two-dimensional spatial information into one-dimensional temporal pulsed streams by leveraging high intermodal dispersion in a multimode fiber. Neural networks are trained to reconstruct images from the temporal waveforms. It can not only detect content-aware images with high quality, but also detect images of different kinds from the training images with slightly reduced quality. The fiber probe can detect micron-scale objects with a high frame rate (15.4 Mfps) and large frame depth (10,000). This scheme combines high speeds with high mechanical flexibility and integration and may stimulate future research exploring various phenomena in vivo.

[1] State Key Laboratory of Precision Measurement Technology and Instruments, Department of Precision Instrument, Tsinghua University, Beijing 100084, China. [2] Key Laboratory of Photonic Control Technology, Ministry of Education, Tsinghua University, Beijing 100084, China. ✉email: xiaoqirong@mail.tsinghua.edu.cn

Ultra-high-speed imaging is vital for observing microscopic and transient physical phenomena[1]. To date, silicon-based imaging sensors, including charge-coupled device (CCD) and complementary metal-oxide-semiconductor (CMOS) cameras, have achieved imaging speeds of up to millions of frames per second (fps)[2]. Some advanced systems have also been invented for even faster transient imaging, reaching trillions of fps, including sequentially timed all-optical mapping photography (STAMP)[3], frequency-domain tomography[4], femtosecond time-resolved optical polarimetry[5], and compressed ultrafast spectral photography[6]. These advanced technologies have helped researchers better understand various transient phenomena, such as lattice dynamics[1], hot-electron diffusion[7], the evolution of laser ablation[8], and the production of electronic plasmas[9]. However, in some other fields, especially in vivo applications[10], high-speed detection requires imaging in narrow spaces, for which the emerging fiber-based imaging technology has unique advantages.

In contrast to bulk imaging systems, fiber-based imaging systems feature high mechanical flexibility, compact sizes, and resistance to ambient interference. These features have made fiber-based imaging a competitive candidate for detecting images under special circumstances, for example, in environments with high temperatures, pressures, or radiation levels. Fiber probes can also penetrate deep into narrow spaces for endoscopy, which is essential in fields such as biomedicine[11] and microfluidics[12]. Fiber endoscopy with a high frame rate is especially necessary in some special scenarios. For instance, a fiber probe can be inserted into the cerebral cortex to examine the fast signals of neural activation[13] or used in vivo to observe chemical dynamics in living tissues[14]. In physics and engineering, such probes can also be used for observing transient physical reactions in closed containers[15] or exploring fuel injection dynamics in internal combustion engines.

For currently prevalent fiber-based imaging systems, the basic principle involves analyzing the light fields at the output fiber facet and reconstructing two-dimensional (2D) images using transmission matrix methods and deep learning methods[16–19]. Due to this principle, they must detect different frame fields at a fixed position, which means that they can only use conventional single-sensor cameras (special cameras such as rotating-mirror cameras[20] and framing cameras with higher frame rates are inapplicable). However, the traditional cameras generally require a balance between the imaging speed and the frame depth (number of frames that can be captured in a single shot) due to a limited readout speed from the pixel arrays to memory[2]. To the best of our knowledge, the world's fastest single-sensor camera has a frame rate of 10 Mfps and frame depth of 256 frames[21], which places an upper limit on performance of the current fiber-based systems in high-speed imaging. Moreover, the silicon-based cameras are sensitive only to wavelengths below 1.1 μm[22], which also limits the applications of these systems in longer infrared bands. In addition, free-space optical elements are commonly required in the collection of output fields from the fiber end, which reduces the level of integration and makes these systems susceptible to environmental disturbances.

A single-pixel imaging method, termed serial time-encoded amplified microscopy[23–25], has been proposed to eliminate pixelated sensors by encoding the spatial information of objects into time-domain signals, which requires only a one-pixel detector. Since each optical pulse can carry the information of one image frame, a high frame rate can be achieved by recording the temporal signals of a pulse train with a high repetition rate. Moreover, the use of one-pixel detectors, such as InGaAs photodiodes can extend the detection wavelengths to longer infrared bands. However, such systems require bulk spatial dispersers, which are not compatible with fiber endoscopy.

Here, we combine the advantages of the time-stretching method and fiber endoscopy and propose a one-pixel method to enable all-fiber high-speed detection of images. Using a single multimode fiber (MMF) as the probe, real-time image acquisition with a frame rate of over 15 Mfps and a shutter time of 45.1 ps was experimentally demonstrated, in which 10,000 frames could be recorded in a single shot. We also verified that the maximum frame rate of the system can be further enhanced to 53.5 Mfps. Leveraging the intermodal dispersion effect in an MMF, we transformed 2D spatial information into one-dimensional (1D) time-domain pulsed waveforms. A neural network model was trained to reconstruct images from the temporal waveforms recorded by an ultrafast photodiode connected to the output end of the fiber. In addition, we propose an all-fiber structure by combining a fiber-output pulse laser, a triple-cladding fiber probe, and a side-pump coupler. This scheme enables high levels of integration and system stability.

## Results

**Principles**. The light fields in an MMF can be resolved into a set of orthogonal spatial modes[26] that enable the transmission of spatial information. It has been verified that the information contained in images with $4N$ resolvable features, where $N$ is the number of fiber modes per polarization, can be carried in a single MMF[27]. When light scattered by an object is collected by an MMF, various fiber modes are excited to different degrees. When an ultrafast pulse laser is used as the illumination source, the energy of each pulse entering the MMF can be dispersed into different modes. Because the different modes have different group velocities, the pulses in these modes will arrive at the opposite end of the MMF with different time delays. If the intermodal dispersion of the MMF is sufficiently large, after transmission through the MMF, a pulse with a temporal duration of less than the delay difference between different modes will be split into a number of isolated subpulses in the time domain, as schematically shown in Fig. 1. If the power of the pulse is sufficiently low and its wavelength bandwidth is sufficiently narrow, both the chromatic dispersion and nonlinear effects in the MMF can be ignored, resulting in the pulse evolution being dominated by intermodal dispersion[28,29] (see Supplementary Note 1 for details). Therefore, the temporal distribution of the train of subpulses depends on the mode composition of the original pulse, which is determined by the spatial distribution of the object. Hence, the spatial information of objects can be encoded into the time waveforms of the output pulses.

**Experimental setup**. The structure of the system is illustrated in Fig. 2a. The illumination pulses from a mode-locked fiber laser are directly coupled into the fiber probe by a side-pump

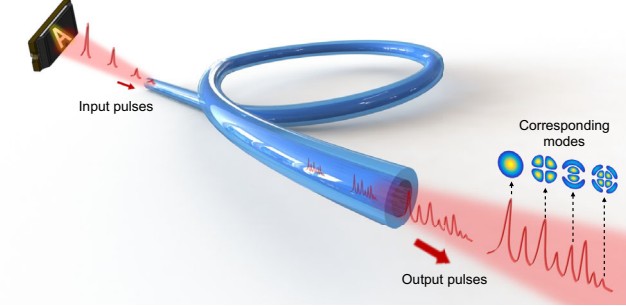

**Fig. 1 Evolution of ultrashort pulses.** The pulses entering the MMF will be split into lots of subpulses due to the large intermodal dispersion in the MMF. Each subpulse contains the energy of a certain mode in principle.

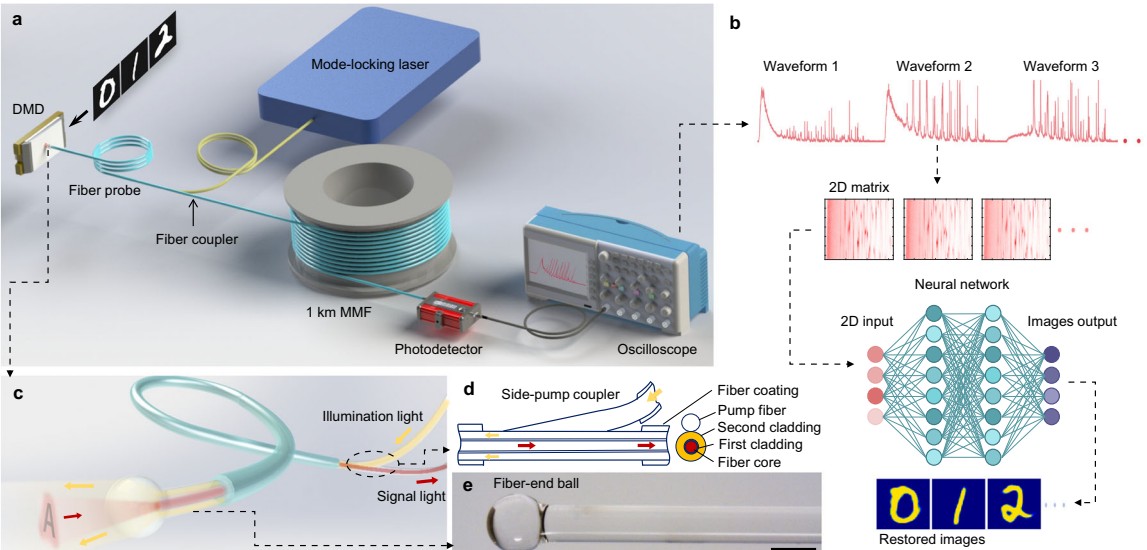

**Fig. 2 Experimental layout. a** Schematic of the experimental setup. **b** Flow of the reconstruction process from waveforms to images. **c** Schematic of the fiber-end ball, the side-pump fiber coupler, and the flow of illumination light (orange arrows) and signal light (red arrows). **d** Structures of the side-pump coupler and triple-cladding fiber probe, where illumination light (orange arrows) is coupled into and transmitted through the cladding of the fiber, while signal light (red arrows) is collected and transmitted through the fiber core in the opposite direction. The cross section of this coupling region is also shown. The triple-cladding fiber consists of the fiber core (red), first cladding layer (blue), and second cladding layer (orange). **e** Micrograph of the fiber-end ball. Scale bar: 500 μm.

coupler[30]. After approximately 2 m of transmission, the illumination pulses emerge from the fiber probe to illuminate the intensity patterns displayed by a digital micromirror device (DMD). The pulse laser operates at a wavelength of 1064 nm with a 3 dB bandwidth of 0.14 nm. While the full width at half maximum of the output pulses is 26.4 ps, it broadens to 45.1 ps after the pulses transmit through the fiber probe and emerge from the fiber-end-ball (see Supplementary Note 1). Then, the light reflected from the patterns reenters the fiber probe, as shown in Fig. 2c. In this way, the illumination and reception of light are integrated into a single fiber probe. The other end of the fiber probe is spliced with a 1 km MMF (50/125 μm, numerical aperture (NA) = 0.22), in which the spatial information carried in the signal pulses is transformed into temporal waveforms. The step-index core of the MMF can provide much greater intermodal dispersion than a graded-index core[31]. In such a long MMF, the delay differences between different modes are sufficiently large to cause each signal pulse to split into a burst of subpulses. The temporal waveforms of the pulses at the other end of the MMF are detected by an ultrafast InGaAs photodetector (spectral response 750–1650 nm, bandwidth 30 GHz) and instantly stored in the memory of an oscilloscope (100 G samples/s.). In the training stage, different displayed images and the corresponding waveforms are used to train the neural network model. After training, the network is capable of recovering new images directly from the acquired waveforms, as shown in Fig. 2b.

The fiber probe is a triple-cladding fiber. The diameters of its core, first cladding, and second cladding are 50, 70, and 360 μm respectively. The core is step-index with an NA of 0.2, and the second cladding has an NA of 0.46. Both the core and the second cladding layer can transmit light. The structure of the side-pump coupler, where the illumination light is coupled into the second cladding layer of the fiber probe (see Supplementary Note 2 for detailed structure), is schematically shown in Fig. 2d. Although the light reflected by the DMD enters both the core and cladding of the fiber probe, only the light in the core (signal light) can enter the MMF due to the matching NA and diameter between the fiber probe core and the MMF core. The end part of the fiber probe is

fused into a microball with a 580 μm diameter, as shown in Fig. 2e, which serves to produce more uniform and focused illumination (in the absence of this microball, the beam emerging from the cladding of the fiber would have an annular shape). This probe can be directly moved very close to microscale objects for imaging, with no requirement of objectives that are vital for conventional cameras. To demonstrate this, the fiber-end ball probe was placed very close to the surface of the DMD such that it could only receive light returning from a very small region of the DMD. The area of this small region measured approximately $200 \times 200$ μm², in which images of approximately $28 \times 28$ pixels could be displayed.

**Image recovery**. Figure 3 shows several example images from the MNIST dataset[32] and their corresponding temporal waveforms. We see that after transmission through the long MMF, a single input pulse splits into a burst of subpulses spanning approximately 45 ns (see Supplementary Note 3 for more waveform details). A U-Net model was trained on 19,000 waveform/image pairs to learn the corresponding mapping. Using the trained model, we could directly recover other new images from the corresponding acquired waveforms. The recovery results corresponding to these example images are shown in the right side of Fig. 3. The results for 1000 test images showed an average fidelity (calculated as the 2D correlation) of 81.8% and an average structural similarity index measure (SSIM, which correlates well with human perception) of 0.78. Compared with previous fiber endoscopy technologies, which generally operate at low frame rates[16,33,34], our scheme showed comparable performance in terms of image quality.

We also tested the reconstruction performance for several different types of images, including handwritten letters from the EMNIST dataset[35] and patterns of clothes from the Fashion-MNIST dataset[36]. After similar training processes, some examples are shown in Fig. 4a, along with the average fidelities and SSIMs. The results indicate high practicability of our scheme. While one waveform corresponds to one image and there is no mutual

interference between neighboring waveforms (see Fig. 4c), the successive pulses can enable detection of images at a frame rate of 15.4 Mfps, which is consistent with the repetition rate of the pulse source. Moreover, the shutter time of the system is equal to the time duration of a single pulse irradiating the DMD. Thus, the shutter time can be as low as 45.1 ps, consistent with the pulse width.

Additionally, the MMF length of 1 km can be further reduced without significantly deteriorating the system performance. The recovery quality under different MMF lengths in the system is

shown in Fig. 4b. The experimental processes using different MMF lengths were consistent with those for the 1 km MMF described above and the same number of digit images were used. There was almost no loss of fidelity as the MMF length was reduced from 1 km to 400 m. After the length was reduced to below 150 m, the image quality deteriorated obviously, indicating that such lengths were too short to split the pulse adequately for separating the information in different modes. In addition, when the MMF length was reduced from 1000 to 400 m, the width of the waveforms was much compressed as shown in Fig. 4c, d. For the 400 m length, a single waveform was much narrow than the period of the pulses, indicating that the temporal space was not fully utilized. Thus, the frame rate could be further increased by lowering the pulse period. Because a single waveform had a length of 18.7 ns in this case, the pulse repetition rate could be increased to 53.5 MHz without any overlap between the neighboring waveforms as shown in Fig. 4d. Thus, it is feasible to increase the frame rate of the current system to 53.5 Mfps by changing the repetition rate of the illumination pulses. Furthermore, if the system is modified to detect a larger image with more pixels, it would require an MMF with more modes. In this case, a larger modal dispersion is required that makes the waveforms becoming broader so that the pulses can be split adequately for separating the information in more modes. However, the broadening of the waveforms will cause the reduction of the frame rate. Thus, considering that the number of modes is in direct proportion to the number of resolvable pixels in the images, the frame rate will be approximately inversely proportional to the number of resolvable pixels.

In the imaging experiments discussed above, the detected images are of the same type as the images used to train the network. Here, we verified that this system could also recover images of different types. To validate this, we replaced the U-Net network with a fully connected network. We found that for the current system, although the U-Net model could realize high-quality imaging of type-aware objects, its ability to image random types of objects, i.e., generalization ability, was not as high as that of the fully connected network (see Supplementary Note 4 for

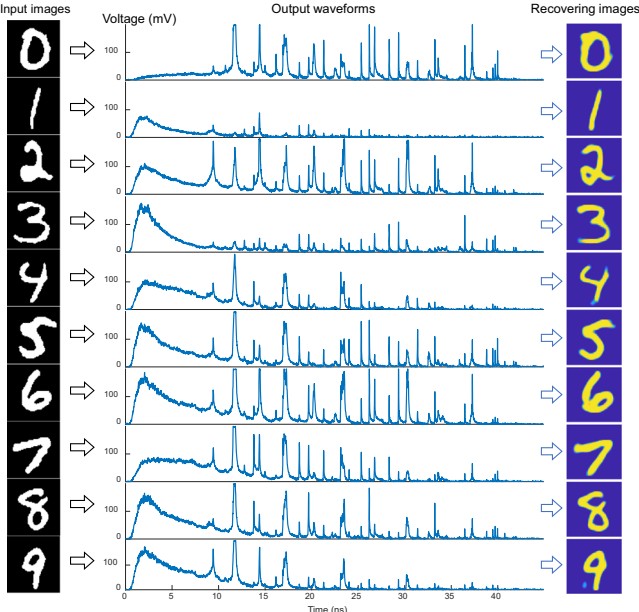

**Fig. 3 Reconstruction process.** The input images of digits 0–9 selected from the test database are transformed into output waveforms. Then the images are recovered from the waveforms.

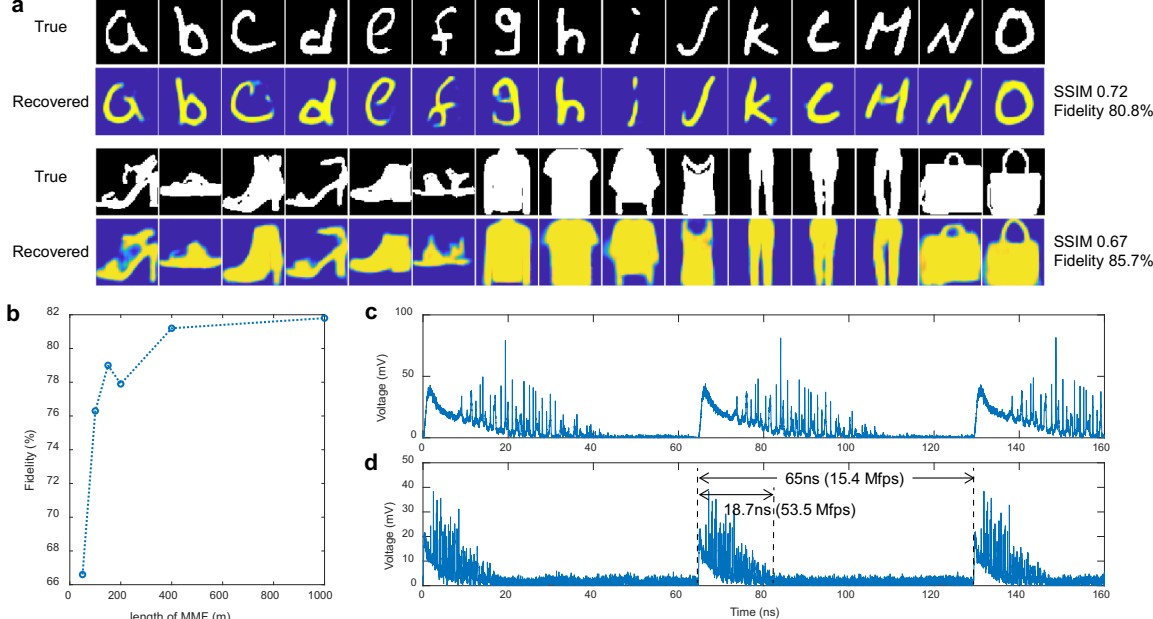

**Fig. 4 Imaging performance analysis. a** Example images of the letters and clothes and the recovered images. **b** Average fidelities when using different MMF lengths in the system. Collected time signals when the MMF length is **c** 1000 m and **d** 400 m respectively. The period of the laser pulses is 65 ns, corresponding to a repetition rate of 15.4 MHz. The width of the waveforms is 18.7 ns, which allows the repetition rate to be further increased to 53.5 MHz.

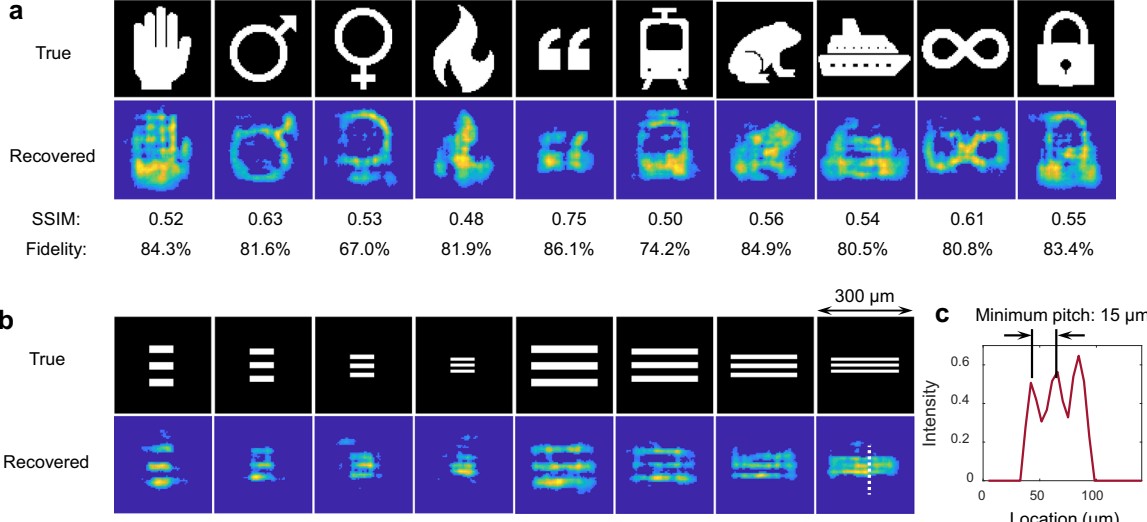

**Fig. 5 Imaging of random types of objects. a** Several recovered images of different types. **b** Recovering results of images containing resolution targets similar to the USAF 1951. The width of the images displayed on the DMD is approximately 300 μm. **c** Vertical intensity profile of the smallest recovered bars along the white dotted line in (**b**). The pitch of 15 μm can be distinguished.

details). Moreover, given that the generalization of a trained neural network is closely connected to the complexity of the training data, we used the images from the Omniglot dataset[37] for training. These images contain patterns made up of nearly random lines. In addition to the original images of the Omniglot dataset, we also generated some other training images based on the original ones by shifting, rotating, or scaling the original patterns to increase the complexity of the training data. Finally, 20,000 images with different complex patterns were used for the training. Then, the trained model was used to restore images of completely different types from the training images. Exemplary results are shown in Fig. 5a and we see that the test images could be recovered with high fidelities. Furthermore, we also verified that this system can be used for detecting grayscale images (see Supplementary Note 9 for details).

Next, we used the same trained model to test the spatial resolution of the system by recovering images of resolution targets similar to the USAF 1951, which contains white bars with different pitches. Here, we adjusted the location of the fiber-end-ball relative to the DMD until it could receive light from a larger region of the DMD surface. The area of this region measured approximately $300 \times 300$ μm$^2$ and included $40 \times 40$ pixels (pixel size of 7.56 μm), which was larger than the previous $28 \times 28$ pixels and thus could help explore the minimum resolution of the system. The recovered results are shown in Fig. 5b, indicating that the smallest bars with pitches of 15 μm (occupying two pixels on the DMD surface) could be distinguished, which is shown more clearer in Fig. 5c.

This system can also be used for high-speed classification, which has great value in fields such as microfluidics[12]. We tested this ability via the classification of handwritten digits based on the acquired waveforms (see Supplementary Note 5 for details). A high accuracy of 91.5% was achieved. We note that image detection through such long fibers has been a major challenge for conventional multimode imaging systems[34] because the disturbance grows more severe as the fiber length increases[33], making the recovery more difficult (the accuracy of digits classification is less than 70% for the speckle-based imaging through a 1 km MMF). However, in our scheme, the classification accuracy remains at such a high level under the same length, indicating high interference immunity and practicability. This superiority can probably be attributed to low crosstalk between different modes when the pulse energy in the modes is separated after transmission over a certain distance in the MMF, thus, the energy coupling between different modes is suppressed. This feature makes our scheme suitable for long-distance detection.

**High-speed detection**. To verify the feasibility of high-speed detection, we adjusted the time scale of the oscilloscope to the maximum (625 μs), allowing it to store approximately 10,000 waveforms in a single record. Although the highest refresh rate of the DMD used here is limited to 4.3 kHz, preventing it from displaying an ultrahigh-speed video that matches our detection frame rate of 15.4 Mfps, the refresh processes when the DMD switches from one image to another are nearly transient and spend only 3 μs (see the recorded waveforms in Fig. 6b). Thus, we chose to detect this refresh process using our system to reveal the detailed process over such a short time. We set the DMD to periodically display two images and simultaneously record the time signals, as shown in Fig. 6b. The detailed waveforms corresponding to one refresh process (marked with a black circle) are shown in Fig. 6a, where we can see the process of the waveforms corresponding to the image 3 gradually changing to the waveforms corresponding to the image 0 within 3 μs. The retrieved successive frames are shown in the insets (a1–a17), from which we can understand the refresh process of the DMD. The whole refresh process can be divided into three stages. In stage 1 (insets a1–a5), the DMD initially displays the image 3, which means that the micromirrors in regions (i) and (ii) of the DMD (see Fig. 6e) are in the on state, while the others are in the off state. The states of the micromirrors are explained in Fig. 6d, where region (ii) represents the overlap between the patterns of 0 and 3. When the DMD starts to refresh to the image 0, the micromirrors in regions (i) and (iii) rotate in opposite directions[38], causing the light in region (i) to fade away. In stage 2 (insets a6–a14), only the light from region (ii) can be observed because the micromirrors there maintain in the on state. In stage 3 (insets a15–a17), the light from region (iii) appears, indicating that the corresponding micromirrors have rotated into on state. Thus, the image 3 has been refreshed to the image 0. For comparison, we also used a commercial high-speed camera to record the refresh process (see the "Methods" section for details), and the real images captured are shown in Fig. 6c. We can see that the change process of the

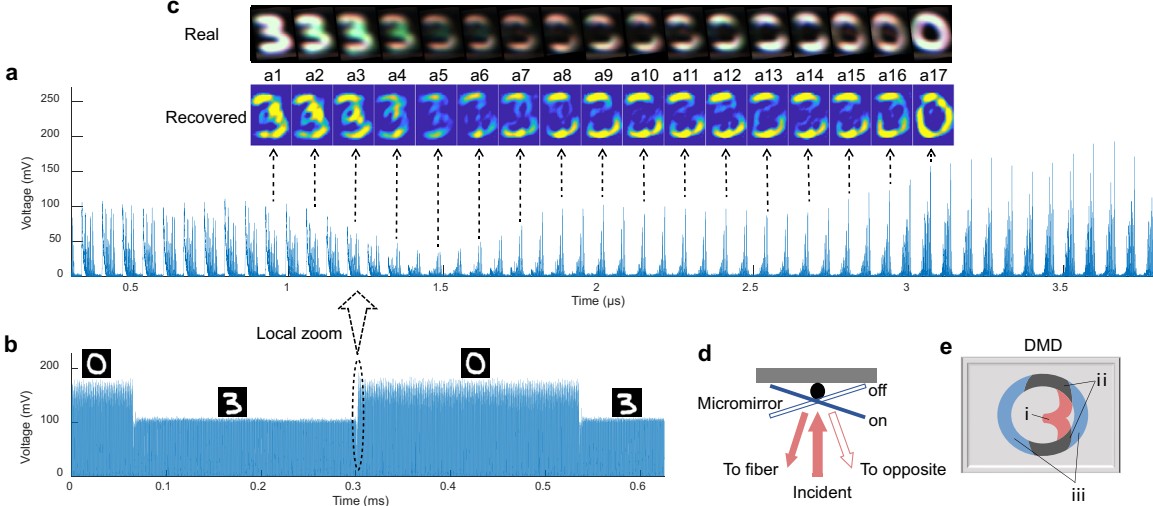

**Fig. 6 Ultra-high-speed imaging. a** Waveforms collected during the transient time when the DMD refreshes from an image 3 to an image 0. Insets (**a1**–**a17**) show the images reconstructed from some of these waveforms. **b** Waveforms recorded with the DMD periodically displaying two images, from which we see that each image has an exposure time of 235 µs, which is used by the DMD to load the data of the next image into the memory cells beneath the pixels. After all data have been loaded, the DMD switches to the other image within only 3 µs, as marked by the dashed circle. The details of this region are shown in (**a**). **c** Real images of the DMD captured during the refresh process using a commercial high-speed camera. **d** Structure of one pixel of the DMD. A DMD chip has many micromirrors on its surface, which correspond to the pixels in the image to be displayed. These micromirrors can be individually rotated by approximately ±12° to an on or off state. In the on state, the illumination light is reflected back to the fiber probe. In the off state, the light is directed in another direction. Thus, the array of micromirrors can produce intensity modulations on the light field. When the DMD refreshes its current image, all micromirrors move to their assigned states at the same moment. **e** Schematic of the patterns of digits 3 and 0 displayed on the DMD chip. When the DMD switches from 3 to 0, the micromirrors in the overlapping region of the two patterns (region ii) do not change their states, while those in regions (i) and (iii) rotate in the opposite directions.

DMD patterns is consistent with what observed using the proposed system.

**System robustness**. To analyze the robustness of the system, we investigated the influence of temperature and fiber bending on the imaging performance. For temperature effect, we changed the environmental temperature from 23 to 27 °C by adjusting the air conditioners. We tested the imaging performance at different temperatures in two different cases. In the first case, the neural network was trained with image/waveform pairs collected with the temperature fixed at approximately 25 °C. In the other case, the network was trained with the data collected at different temperatures, called joint training. The details of the experiment and test results are shown in Supplementary Note 6. While the average fidelity of the recovered images remains above 70% within a temperature variation of 0.5 °C in the first case, this variation range increases to 3 °C for the joint training case. This temperature sensibility is mainly caused by the temperature-induced index-distribution change in the MMF which will influence the temporal distribution of the subpulses. To investigate the bending effect, we fixed the fiber-end ball and bent the fiber probe into a semicircle with variable radii as shown in the inset of Supplementary Fig. 11b. Similarly, we tested the imaging performance under different bending states in two cases: training with the data collected under one bending state and joint training with the data collected under different bending states. The results (see Supplementary Fig. 11) show that in the first case, the average fidelity can remain above 70% when the bending radius changes from 28 to 22 cm. And for the joints training, the 70% fidelity is obtained in the range of 28–19 cm, showing a higher robustness. The sensibility to the bending is mainly caused by the fiber-stress-induced modal crosstalk inside the MMF. In summary, we verified that this system has certain robustness for practical applications.

## Discussion

Because the proposed scheme requires only a single photodiode rather than pixelated sensors, it can be easily applied to other wavelengths. For example, considering that the InGaAs-based photodiode used here has high sensitivity over a broad band from 1 to 1.6 µm[39], while the silica fiber has very low attenuation in this band, our scheme can be easily extended to other wavelengths within this band. This will be highly valuable for real applications because conventional Si-based CCD and CMOS cameras are sensitive only to wavelengths below 1.1 µm[22]. Our method also has the potential to operate in the mid-infrared or THz bands, considering the development of photodetectors in these bands. In the mid-infrared band, the use of a fluoride-glass fiber can significantly reduce optical loss, which makes it possible to develop long waveguides with high intermodal dispersion in these bands. In addition, we can see from Fig. 4b that the MMF length may be potentially reduced to 150 m with little degradation of image quality. The required MMF length can be further reduced by increasing the NA of the MMF, which increases the intermodal dispersion. Thus, although current technology can only fabricate fluoride-glass fibers with losses on the order of 0.1 dB/m[40], the total loss can be controlled to an acceptable range. In addition, there is still room to lower the loss in fluoride-glass fibers according to the theoretical predictions of Shibata et al.[41]. Thus, it is possible to extend the wavelength of this system to the mid-infrared region. Also, in the THz band, much work has focused on the development of waveguides with reduced loss and dispersion[42], which provides a certain possibility to apply the method to this band. This will be helpful for detecting certain materials that have strong responses only at these wavelengths or for detection under special conditions in which only light in these bands can be transmitted with low loss. In summary, our scheme offers an alternative approach for observing vivid physical phenomena in a vast number of scenarios.

The performance of our demonstrated proof-of-principle system may be further improved. The wavelength of the source used here (1064 nm) is much longer than those adopted in most previous studies[16,26,33,34], which resulted in a much smaller number of excited modes and, thus, much less spatial information carried in the MMF. Hence, upon using an MMF with a larger core and higher NA, more spatial information can be collected, and the resolution of the recovered images will be much higher. In addition, the shutter time can be further shortened to enable the detection of faster events by using shorter pulses. More importantly, with a fiber amplifier spliced to the end of the MMF, the pulse signals can be significantly amplified, which will greatly enhance the sensitivity of the detection system to make it suitable for detecting very weak signals. Moreover, because the illumination zone and intensity of the applied fiber probe are limited, the current system is only suitable for detecting small objects. For larger-object detection, an objective can be used in front of the fiber probe to couple more light from the object into the probe. Additionally, for brighter illumination, auxiliary illumination can be adopted as discussed in Supplementary Note 7.

Our scheme can be further modified to detect 3D objects by combining it with the existing time-of-flight technique[43–45], in which ultrafast pulses are generally used to illuminate objects of interest and an ultrafast camera is used to detect the reflected light at different arrival times. Because the light reflected from different depths on the object will arrive at the camera with different time delays, the variations in 2D images captured over time can reveal the 3D information of the object. The system presented in this paper is naturally compatible with the time-of-flight method because we also adopt an ultrafast pulse laser for illumination. If the fiber probe is used to detect a 3D object, the temporal waveforms will contain both depth information and 2D spatial information. Thus, the use of specific reconstruction algorithms will make it possible to recover the 3D information encoded in these ultrafast time signals.

## Methods

**Experiments**. The laser source is a homemade Yb-doped mode-locked fiber laser with an average output power of approximately 1 W. The fiber probe is a triple-cladding fiber with diameters of 50/70/360 μm (NUFERN FUD-4658, BD-S50/70/360-22FA-HP). The homemade fiber coupler couples light from the source into the second cladding layer of the probe. The fiber-end ball at the end of the probe was produced via fusion with a fusion splicer. Because we adopted a pulse laser, considering that the DMD can only perform grayscale modulation on continuous light, all images were binarized before loading into the DMD. The DMD (Texas Instruments DLP4500) consists of 912 × 1140 micromirrors, each being 7.56 μm in size. The photodetector (Thorlabs DXM30BF) has a 30 GHz response bandwidth and a 15 ps impulse response with a sensible spectrum of 750–1650 nm. The photodetector receives light through an OM4 (50/125 μm) fiber, which is connected to the other end of the MMF. The oscilloscope (Tektronix MSO73304DX) has a 33 GHz analog bandwidth and a sample rate of 100 G/s. Its maximum record length is 62.5 Mega samples.

During the process of collecting waveforms of the training images, we set the oscilloscope to automatically save waveforms at a speed of 4 waveforms per second. We have verified that training with 10,000 samples was adequate for the network to achieve the optimal performance (see Supplementary Note 8). Thus, the whole collection procedure spanned 42 min. The collected raw data was processed before fed to the neural network. Because a recorded time signal included several periods of pulses, one period would be selected and extracted as one waveform. Finally, the waveforms of all images were put together and converted into a matrix data. This was processed by a MATLAB program, requiring approximately 5 min. Training the U-Net network with 10,000 sample data required approximately 4 min. We used the online computing resource from Google Colab that provides a Tesla P100-PCIE GPU. In conclusion, the whole calibration process, including sample collecting, processing, and training, required approximately 51 min.

The high-speed camera (MotionBLITZ EoSens® mini) used in the high-speed imaging experiment has a frame rate of 40 kfps, which is too slow to record a refresh process of the DMD in real time. Thus, the images showing this transient process in Fig. 6c were actually not captured during a single refresh process. Instead, they were obtained using the following method. First, we acquired a large number of images while the DMD periodically switched between two images. Because the time of a single refresh process occupies only a very small part of the

switching period, as shown in Fig. 6b, only a small number of images were captured exactly during the refresh processes. Because these images tended to record different states of the process, they could be combined to present a continuous refresh process. The exposure time of the camera was set to the minimum to capture these transient states.

**Neural networks**. The structure of the U-Net network is shown in Supplementary Note 5. In training, the original images were all interpolated into 64 × 64 matrixes as the output of the network, and the 4096-point waveforms were reshaped into 64 × 64 matrixes as the input. The Fully-Connected network consists of five fully connected layers, which are one dimensional, and thus the matrixes of images should be reshaped into vectors and thus the reshaping of the waveforms is not required.

## Data availability

The image and waveform data that are necessary to evaluate the conclusions in this study are available in the Tsinghua cloud [https://cloud.tsinghua.edu.cn/f/f7e530af7c6c44caaf74/?dl=1].

## Code availability

The python codes used in this study are available in the Tsinghua cloud [https://cloud.tsinghua.edu.cn/f/f7e530af7c6c44caaf74/?dl=1].

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

## Acknowledgements

Q.X. acknowledges financial support from National Natural Science Foundation of China (Grants No. 62122040, 62075113, and 61875103). We thank Dr. Zeyi Li and Dr. Yvze Lu for guidance in neural-network algorithms.

## Author contributions

Z.L. and Q.X. conceived and performed the most experiments and calculations. L.W. (Lele Wang) made the schematic diagrams. T.H., Y.Y. and L.W. (Liuyue Wang). contributed to the deep-learning algorithms. Q.L., M.G. and P.Y. discussed the results and contributed to the writing of the paper. J.T., D.L., S.H. and Y.M. contributed to part of the experiments. Q.X. conceived and led the project.

## Competing interests

The authors declare no competing interests.
