## [Peer Review File · Nature Communications]

All-fiber high-speed image detection enabled by deep learningREVIEWER COMMENTS

Reviewer #1 (Remarks to the Author):

The authors present an interesting method that combines neural networks and modal dispersion in a multimode fiber to obtain imaging at a fast frame rate.

The manuscript is well written and clear to understand. The idea is elegant. The ultra fast detection scheme should be toned down as it is not convincingly superior to current high speed cameras. The idea is nonetheless elegant and could be applied to other wavelength ranges and hence this work could be acceptable in Nature Communication.

The authors write "The basic principle of these technologies is to analyze the light fields at the output fiber facet and reconstruct the 2D images using such as transmission matrix methods and deep learning methods 14,17" Reference 14 is not using a transmission matrix to reconstruct an image but a Grin lens. The authors may reference more relevant works in multimode fibers which use the transmission matrix to produce images such as e.g. Loterie D., Farahi S. Papadopoulos I., Goy A., Psaltis D., Moser C., Digital confocal microscopy through a multimode fiber, Optics Express, 23, 18, 2015.

Indicate the probe core diameter, type of fiber (step index of GRIN), wavelength of the laser directly in the main text of the manuscript.

Indicate the pulse length of the modelocked laser before launching it in the 2m fiber probe and at the distal end (at the DMD).

How is the 15 Mfps computed ? Is it the inverse repetition rate of the laser ?

The frame rate of 15 Mfps sounds impressive, however this frame rate is computed with 28×28 pixels = 784 pixels.

This corresponds to 11.7 Gpixels/sec which is the same data rate as standard commercial fast cameras.

How does the frame rate scales with image size and fiber length ? A larger image would require a fiber with more modes. Assuming the same modal dispersion as the fiber used in the experiments i.e a time per pixel of $50 \text{ ns}/784 \text{ pixels} = 50 \text{ ps/pixel}$,

For a Megapixel, this would give a length of time of 50 us or a $10^5/5=20 \text{ kfps}$, which is of the order of what can be achieved with fast cameras.

The authors rightly mention that the single detector has an advantage in the infrared where cameras are not silicon based.

The authors emphasize biomedical applications and endoscopy in the abstract. The authors mention fluoride fibers as a potential use case, however these fibers have a loss of the order of 0.1 dB/m and therefore, this limits the fiber length to tens of meters which would be too short to produce a temporal spread. Clarifications are here necessary.

Reviewer #2 (Remarks to the Author):

The paper proposes a method of reconstructing images transmitted through a multimode fiber probe from their temporal wavefronts. I find the idea to be very interesting. The paper is sound, thorough, and of large interest to the community.

I'm impressed by the results demonstrated in the manuscript. However, there is not enough detail provided and the description is not really clear.

1. In the abstract, the authors claim "high mechanical flexibility" of their system. However, it's well known that a transmission matrix of a multimode fiber is very sensitive to fiber deformations. How robust is the proposed approach to fiber bending or temperature changes or other external factors? How stable is the system? I guess, it's required to repeat the training procedure if the fiber probe and/or MMF has been moved. Could you discuss it in the manuscript.

2. Could the authors provide more information on the image recovery procedure. How long is the training step? How many waveform/image pairs are needed for different datasets? How much time does it take to recalibrate

the system for a new object type (both measurements and training)?

3. Could the authors provide more information on the experimental setup. What model of a DMD has been used? What photodetector (photodiode) and oscilloscope have been used? Key parameters of all the elements should be provided.

4. As far as I can see, the proposed approach allows to reconstruct the images if it's known what kind of object we are looking at. I would call it content-aware kind of imaging. I think this limitation should be highlighted [in the abstract] and discussed.

5. The authors demonstrate only binary (black-and-white) images. Does the proposed approach allow to image more complex gray scale objects?

6. Please discuss the spatial resolution achieved. What was the average feature size of the reconstructed images?

Detailed Responses to the Reviewers

We sincerely thank the reviewers for the constructive comments. The suggestions were well taken.

According to these suggestions, we have supplemented abundant experiments and analyses regarding the imaging properties of our system. Our manuscript and the supplementary materials have been carefully revised according to the reviewer's instructions. In the following, we respond to the comments one by one, with every comment listed first and the corresponding response following.

Response to Reviewer #1

Comment 1: The manuscript is well written and clear to understand. The idea is elegant. The ultra fast detection scheme should be toned down as it is not convincingly superior to current high speed cameras. The idea is nonetheless elegant and could be applied to other wavelength ranges and hence this work could be acceptable in Nature Communication.

Response 1:

We would like to sincerely thank the reviewer for carefully assessing our manuscript and raising the constructive comment.

The reviewer's viewpoint on the imaging speed is very reasonable. To our best knowledge, the highest pixel rate of commercial cameras can reach 60 Gpixels/sec according (1280×896 @ 50 kfps, <https://www.nacinc.com/pdf.php?pdf=/datasheets/Memrecam-ACS-1-M60-40-Data-Sheet.pdf>), which could be comparable to that delivered in this manuscript (the maximum pixel rate of our system is 21.4 Gpixels/sec as discussed in **Response 4**). Thus, in terms of data rate (or pixel rate), our approach does have a similar performance to fast cameras.

Therefore, we have modified the claim of “ultrafast” to “high-speed” throughout the manuscript. Some of the modifications are shown below (marked as yellow).

1	All-fiber high-speed image detection enabled by deep learning
2	
3	Abstract: Detection of dynamic scenes at ultra-high speeds serves as a foundation for modern
4	engineering, chemistry, material science and biomedicine. In biomedical applications, in vivo
5	microscopic imaging is often required, which has led to the development of fiber-probe-based
6	endoscopy. However, the combination of high-speed image acquisition with fiber endoscopy, which
7	is vital for exploration of transient biomedical phenomena, has not yet been achieved. Here, we
8	propose a scheme of all-fiber image detection at a high speed without any free-space optical

However, we emphasize that, in terms of imaging speed, our system still has certain advantages compared with traditional fast cameras in some conditions, as described below:

- 1) The maximum frame rate of this system that can be achieved is 15.4 Mfps (proved in **Response 4**), which is much higher than the fast cameras. Some representative cameras are shown in the table below:

Model	Highest frame rate	Resolution @ the highest fps
ix-cameras i-SPEED 727	2.45 Mfps	280×12
DITECT HAS-D71	0.12 Mfps	640×12
MEMRECAM HX-3e	0.22 Mfps	320×96
PHANTOM v2512	1 Mfps	128×32
FASTCAM SA-Z	2.1 Mfps	128×8
Memrecam ACS-1-M60	1 Mfps	1280×32

- 2) We can also see from the table above that the pixel areas of the fast camera when they operate at the highest fps generally have long and narrow shapes, which will result in much smaller effective imaging sizes (see discussion in **Response 4**). On the contrary, our approach always has a square imaging area due to the symmetry of the fiber core. Thus, even with similar pixel rate to fast cameras, if just consider the condition of operating at very high frame rates, our approach will have higher effective pixel rate than fast cameras.
- 3) Although the pixel rate of our approach is similar to that of fast cameras when they operate at relatively low frame rate, their pixel rates are predicted to continuously decrease as the operating frame rate rises (see **Response 4** for more details) due to the limited data read speed of the CCD or CMOS chip. When operating at >1 Mfps, the pixel rates of fast camera are predicted to be lower than our approach and this advantage will increase as the frame rate goes higher.

Comment 2: The authors write “The basic principle of these technologies is to analyze the light fields at the output fiber facet and reconstruct the 2D images using such as transmission matrix methods and deep learning methods 14,17”. Reference 14 is not using a transmission matrix to reconstruct an image but a Grin lens. The authors may reference more relevant works in multimode fibers which use the transmission matrix to produce images such as e.g. Loterie D., Farahi S. Papadopoulos I., Goy A., Psaltis D., Moser C., Digital confocal microscopy through a multimode fiber, Optics Express, 23, 18 , 2015.

Response 2:

We thank the reviewer for this constructive suggestion and the kind reminder. We found the abovementioned references very interesting and helpful.

Therefore, we have added the following three representative references that used transmission-matrix methods:

- [1] Loterie, D. et al. Digital confocal microscopy through a multimode fiber. Optics Express 23 (2015).
- [2] Choi, Y., Yoon, C., Kim, M., Yang, T. D. & Choi, W. Scanner-Free and Wide-Field Endoscopic Imaging by Using a Single Multimode Optical Fiber. Physical Review Letters 109, 203901 (2012).
- [3] Caramazza, P., Moran, O., Murray-Smith, R. & Faccio, D. Transmission of natural scene images through a multimode fibre. Nature Communications (2019).

The modification in the manuscript:

47 For currently prevalent fiber-based imaging systems, the basic principle involves analyzing the light
48 fields at the output fiber facet and reconstructing two-dimensional (2D) images using transmission
49 matrix methods and deep learning methods¹⁷⁻²⁰. Due to this principle, they must detect different
50 frame fields at a fixed position, which means that they can only use conventional single-sensor

440 17 Rahmani, B., Loterie, D., Konstantinou, G., Psaltis, D. & Moser, C. Multimode optical fiber
441 transmission with a deep learning network. *Light: Science & Applications* **7**, 69 (2018).¹⁷
442 18 Loterie, D. *et al.* Digital confocal microscopy through a multimode fiber. *Optics Express* **23** (2015);
443 19 Choi, Y., Yoon, C., Kim, M., Yang, T. D. & Choi, W. Scanner-Free and Wide-Field Endoscopic
444 Imaging by Using a Single Multimode Optical Fiber. *Physical Review Letters* **109**, 203901 (2012);
445 20 Caramazza, P., Moran, O., Murray-Smith, R. & Faccio, D. Transmission of natural scene images
446 through a multimode fibre. *Nature Communications* (2019).^{18,19,20}

Comment 3: Indicate the probe core diameter, type of fiber (step index of GRIN), wavelength of the laser directly in the main text of the manuscript. Indicate the pulse length of the mode-locked laser before launching it in the 2m fiber probe and at the distal end (at the DMD).

Response 3:

Thank you for the reminder. The suggestion is taken fully. These parameters are truly very important.

The details of the fibers and laser source are as follows:

The fiber detection system consists of two different fibers. The 2 m fiber probe (NUFERN FUD-4658, BD-S50/70/360-22FA-HP) is a triple-cladding fiber. The diameters of its core, first cladding and second cladding are 50 μm , 70 μm and 360 μm , respectively. The core is a step-index type and has an NA of 0.2. The multimode fiber (MMF), which provides modal dispersion, also has a step-index core with an NA = 0.22, and the diameters of the core and cladding are 50 μm and 125 μm , respectively.

The mode-locking pulse laser operates at a wavelength of 1064 nm with a 3 dB bandwidth of 0.14 nm as shown in Fig. R1. The full width at half maximum of the output pulse from the final amplification stage is 26.4 ps. However, as the reviewer noted, this pulse width broadens to 45.1 ps after the pulses are transmitted through the fiber probe and emerge from the fiber-end ball, as shown in Fig. R2. This broadening is caused by the slight modal dispersion in the fiber probe. However, this slight broadening has little influence on the pulse spread because the long MMF can provide sufficiently high modal dispersion. If required, this broadening effect can be relieved by replacing the cladding pump method with the core pump method, which couples the illumination light into the core of the fiber probe. Because the modal dispersion in the core is lower than that in the cladding with an NA of 0.46 in this method, pulse broadening is greatly restrained.

Fig. R1 Measured spectrum of the pulse source.

Fig. R2 Measured pulse shapes before the pulses enter the MMF and after they emerge from the fiber end.

We have supplemented detailed information directly in the main text:

105 **Experimental setup.** The structure of the system is illustrated in Fig. 2a. The illumination pulses
106 from a mode-locked fiber laser are directly coupled into the fiber probe by a side-pump coupler³¹.
107 After approximately 2 meters of transmission, the illumination pulses emerge from the fiber probe
108 to illuminate the intensity patterns displayed by a digital micromirror device (DMD). The pulse laser
109 operates at a wavelength of 1064 nm with a 3 dB bandwidth of 0.14 nm. While the full width at half
110 maximum of the output pulse is 26.4 ps, it broadens to 45.1 ps after the pulses transmit through the
111 fiber probe and emerge from the fiber-end-ball (see Supplementary Note 1). Then, the light reflected
112 from the patterns reenters the fiber probe, as shown in Fig. 2c. In this way, the illumination and
113 reception of light are integrated into a single fiber probe. The other end of the fiber probe is spliced
114 with a 1-km MMF (50/125 μm , numerical aperture (NA) = 0.22), in which the spatial information
115 carried in the signal pulses is transformed into temporal waveforms. The step-index core of the
116 MMF can provide much greater intermodal dispersion than a graded-index core³². In such a long
117 MMF, the delay differences between different modes are sufficiently large to cause each signal pulse
118 to split into a burst of subpulses. The temporal waveforms of the pulses at the other end of the MMF
119 are detected by an ultrafast InGaAs photodetector (spectral response 750–1650 nm, bandwidth 30
120 GHz) and instantly stored in the memory of an oscilloscope (100 G samples/sec.). In the training
121 stage, different displayed images and the corresponding waveforms are used to train the neural
122 network model. After training, the network is capable of recovering new images directly from the
123 acquired waveforms, as shown in Fig. 2b.[↵]
124 [↵]
125 The fiber probe is a triple-cladding fiber. The diameters of its core, first cladding and second
126 cladding are 50 μm , 70 μm and 360 μm respectively. The core is the step index with an NA of 0.2,
127 and the second cladding has an NA of 0.46. Both the core and the second cladding layer can transmit

354 **Experiments.** The laser source is a homemade Yb-doped mode-locked fiber laser with a spectral
355 width of 0.2 nm. The average output power is 1 W. The fiber probe is a triple-cladding fiber with
356 dimensions of 50/70/360 μm (NUFERN FUD-4658, BD-S50/70/360-22FA-HP). The NAs of the
357 step-index core and the second cladding layer of the triple-cladding fiber are 0.2 and 0.46,
358 respectively. The homemade fiber coupler couples light from the source into the second cladding
359 layer of the probe. The fiber-end ball at the end of the probe was produced via fusion with a fusion
360 splicer. Because we adopted a pulse laser, considering that the DMD can perform only grayscale
361 modulation of continuous light, all images were binarized before loading into the DMD. The DMD
362 (Texas Instruments DLP4500) consists of 912×1140 micromirrors, each being 7.56 μm in size. The
363 photodetector (Thorlabs DXM30BF) has a 30 GHz response bandwidth and a 15 ps impulse
364 response. The sensible spectrum is 750–1650 nm. The photodetector receives light through an OM4
365 (50/125 μm) fiber, which is connected to the other end of the MMF. The oscilloscope (Tektronix
366 MSO73304DX) has a 33 GHz analog bandwidth and a sample rate of 100 G/s. The maximum record
367 length is 500 Mega samples.[↵]

And the information about the pulses are added to the Supplementary Note 1 of the revised supplementary materials.

Comment 4: How is the 15 Mfps computed? Is it the inverse repetition rate of the laser? The frame rate of 15 Mfps sounds impressive, however this frame rate is computed with 28x28 pixels = 784 pixels. This corresponds to 11.7 Gpixels/sec which is the same data rate as standard commercial fast cameras.

Response 4:

We thank the reviewer for raising this important comment. This comment was well taken. In the following we will carefully compare the advantages of our proposed scheme with commercial fast cameras in terms of the data rate (pixel rate) parameter. The computation/evaluation details for the 15Mfps rate are also further elaborated.

(1) Computation of the frame rate

The frame rate is equal to the repetition rate of the laser in the manuscript due to that every single pulse carries the information of one frame and the different pulse waveforms do not overlap in time as shown in Fig. R3. Because the waveform length is shorter than the pulse period, the frame rate of the current system has the potential to be further improved as the figure shows.

Fig. R3 Detected time signals of the current system.

As shown in Fig. R4, the frame rate can be further increased to 22 Mfps upon adopting a pulse laser with a repetition rate of 22 MHz instead of the original 15 MHz.

Fig. R4 Optimal repetition rate of the laser pulse.

We have modified the description of the frame rate in the manuscript to make it more clear:

164 We also tested the reconstruction performance for several different types of images, including
 165 handwritten letters from the EMNIST dataset³⁶ and patterns of clothes from the Fashion-MNIST
 166 dataset³⁷. After similar training processes, some examples are shown in Fig. 4a, along with the
 167 average fidelities and SSIMs. The results indicate high practicability of our scheme. While one
 168 waveform corresponds to one image and there is no mutual interference between neighboring
 169 waveforms (see Fig. 4c), successive pulses can enable detection of images at a frame rate of 15.4
 170 Mfps, which is consistent with the repetition rate of the pulse source. Moreover, the shutter time of
 171 the system is equal to the time duration of a single pulse irradiating the DMD. Thus, the shutter time
 172 can be as low as 45.1 ps, consistent with the pulse width.⁴¹

(2) Comparisons with the fast cameras

The reviewer has insightfully pointed out that the pixel rate of the system demonstrated in the manuscript is similar to those of commercial fast cameras. Hence, we toned down the statement of

"ultra fast imaging" in the manuscript and instead, we emphasized that the present approach is in the category of high-speed imaging in both the title and the abstract.

Nevertheless, we would like to sincerely clarify that, although the pixel rate of our system is similar to that of fast cameras when they operate at relatively low frequencies, our approach truly have several advantages over fast cameras when operating at extremely high frame rates.

First of all, let us discuss about the maximum pixel rate that our system can reach.

As discussed above, the maximal frame rate depends on how much the pulse spreads. In the original experiment, we used a 1 km MMF and the pulses spread by approximately 45 ns. Considering that the resolution of the system is approximately 20×20 (may also see **Response 6** to the reviewer #2), the pixel rate of the current system is 20×20 pixels/45 ns = 8.9 Gpixels/sec. We emphasize that this value is not the maximum pixel rate that our system can reach. This is because that the pulse spreading over 45 ns is not necessary, and which can be much compressed with negligible influence on the imaging performance. To explore the maximum pixel rate, we reduced the spreading length by using shorter MMFs and examined the imaging performance. The waveforms corresponding to different MMF lengths are shown in Fig. R5. Although the waveforms were compressed as the MMF length decreased, the details of the peaks in the waveforms were retained. In these experiments, all devices including the source, photodetector and oscilloscope are unchanged except for the MMF.

Fig. R5 Waveforms when different MMF lengths were used.

The quality of the restored images versus the length of the MMF is plotted in Fig. R6. The fidelity remains nearly unchanged when the length of the MMF is reduced to 400 m.

Fig. R6 System performance when different MMF lengths were used..

Example recovered results are also shown in Fig. R7. We see that the images could be well restored when the MMF length was reduced to 400 m, which corresponded to a pulse spreading of 18.7 ns. Thus, the frame rate of the system can be further increased to $1/18.7 \text{ ns}=53.5 \text{ Mfps}$ by adopting a pulse laser with a repetition rate of 53.5 MHz. Although we did not truly adopt such a pulse laser in the system due to our laboratory conditions, this approach is obviously feasible. Hence, the maximal pixel rate can reach $20 \times 20 \times 53.5 \text{ MHz}=21.4 \text{ Gpixels/sec}$. The maximum length of the MMF can be further reduced by adopting shorter pulses and faster photodetectors (both of which are commercially available), which would yield even higher pixel rates.

Fig. R7 Example recovered images when different MMF lengths were used.

Next, we investigated several cameras approximately representing the highest frame rates on the market and calculated their pixel rates when operated at different frame rates as shown in Fig. R8. We found that the pixel rates of these cameras generally decrease with their operating frame rates, which is probably caused by the limited data-read-out speed from the CMOS chips. When the frame rate reaches approximately 1 Mfps, their pixel rates decrease to approximately 20 Gpixels/sec. It can be easily predicted that their pixel rates will continuously decrease when operating at even higher frame rates. Considering that our system can achieve a pixel rate of 21.4 Gpixels/sec when operating at the frame rate of 53.5 MHz, if those fast cameras are set to operate at this high frame rate, their pixel rates are predicted to be much lower than ours.

Fig. R8 Pixel rates of several representative fast cameras when operating at different frame rates. The red arrows indicate the pixel areas when the cameras are operating at the highest frame rates.

Moreover, when the fast cameras operate at their highest frame rates, we note that their pixel areas generally have long and narrow shapes (as indicated by the red arrows in Fig. R8). Thus, in practical, their image quality would be curbed by the small horizontal resolutions. For example, when the MEMRECAM ACS-1 M60 camera operates at the frame rate of 1 Mfps, although the pixel rate is very large and similar to that of ours, it can only detect images of 1280×16 pixels: even though the vertical resolution is large, it only has a horizontal resolution of 16 pixels. Thus, for observed objects that have square shapes (which is very common in practical, such as cells), the large vertical resolution of 1280 cannot be fully used, limiting the effective pixel area to only 16×16. Unlike the cameras, our approach has equal resolutions in both the horizontal and vertical directions due to the symmetrical shape of the fiber. From this point of view, our approach has larger effective pixel rates than those cameras.

In summary, our approach has higher pixel rate than fast cameras when operating at extremely high frame rates (> 1 Mfps) and moreover, the detected images of our system have a square shape, which gives a larger effective pixel resolution.

The discussion about the maximum frame rate has been supplemented to the manuscript:

174 Additionally, the MMF length of 1 km can be further reduced without significantly deteriorating the
 175 system performance. The recovery quality under different MMF lengths in the system is shown in
 176 Fig. 4b. The experimental processes using different MMF lengths were consistent with those for the
 177 1-km MMF described above and the same number of digit images were used. There was almost no
 178 loss of fidelity as the MMF length was reduced from 1 km to 400 m. After the length was reduced
 179 to below 150 m, the image quality deteriorated obviously, indicating that such lengths were too short
 180 to split the pulse adequately for separating the information in different modes. In addition, when the
 181 MMF length was reduced from 1000 m to 400 m, the width of the waveforms was much compressed
 182 as shown in Fig. 4c, d. For the 400 m length, a single waveform was much narrower than the period
 183 of the pulses, indicating that the temporal space was not fully utilized. Thus, the frame rate could
 184 be further increased by lowering the pulse period. Because a single waveform had a length of 18.7
 185 ns in this case, the pulse repetition rate could be increased to 53.5 MHz without any overlap between
 186 the neighboring waveforms as shown in Fig. 4d. Thus, it is feasible to increase the frame rate of the
 187 current system to 53.5 Mfps by changing the repetition rate of the illumination pulses. Furthermore,

Fig. 4 (a) Example images of the letters and clothes and the recovered images. (b) Average fidelities when using different MMF lengths in the system. (c, d) Collected time signals when the MMF length is (c) 1000 m and (d) 400 m respectively. The period of the laser pulses and the width of the waveforms are marked in (d), along with the corresponding frequencies calculated by the reciprocals of the time durations.⁴²

Comment 5: How does the frame rate scales with image size and fiber length? A larger image would require a fiber with more modes. Assuming the same modal dispersion as the fiber used in the experiments i.e a time per pixel of 45 ns/784 pixels = 50 ps/pixel. For a Megapixel, this would give a length of time of 50 us or a $10^5/5=20$ kfps, which is of the order of what can be achieved with fast cameras.

Response 5:

We sincerely thank the reviewer for the constructive comment. All of your comments on our manuscript were very accurate and reasonable.

First, we discuss how the frame rate changes with the detected image size and fiber length. To restore the image information from the waveforms, the modal dispersion should be larger than a threshold at which the pulses are adequately spread. In **Response 4**, we found that the minimum required MMF length is 400 m, which corresponds to a time spread of 18.7 ns. Thus, we can conclude that the threshold of the required modal dispersion is approximately 18.7 ns/20×20 pixels = 46.8 ps/pixel, which is the reciprocal of the maximum pixel rate that the system can reach. Hence, the system can operate at the highest pixel rate when the modal dispersion is at the threshold. Because the pixel rate = frame rate × pixel number, the maximum frame rate that can be achieved is inversely proportional to the image size. For example, if we want to transmit an image containing N pixels through the

MMF, the pulse must be spread into a waveform with a length of $46.8 \times N$ ps, which gives a maximum frame rate of $10^{12}/(46.8 \times N)$ fps.

There is no direct relationship between the fiber length and the frame rate. Actually, the fiber length depends on the required number of modes. To detect images of larger sizes, the number of modes in the MMF must be increased proportionally (the maximum number of pixels that the fiber imaging can achieve is proportional to the number of modes, Reza et al., Optics Express, 2013, 21(2)), which can be realized by enlarging the fiber core area. However, enlarging the fiber core has little effect on the total modal dispersion (Gloge D., Applied optics, 1971, 10(10)):

$$\Delta\tau_{gr} = \left(1 - \frac{2}{v}\right) \frac{\Delta n L}{c} \approx \frac{\Delta n L}{c}$$

Thus, to keep the modal dispersion at the threshold of 46.8 ps/pixel, the length of the MMF must be increased nearly proportionately to the number of modes. Thus, the fiber length should be nearly directly proportional to the image size.

We have added some discussion about these relationships in the manuscript:

186 the neighboring waveforms as shown in Fig. 4d. Thus, it is feasible to increase the frame rate of the
 187 current system to 53.5 Mfps by changing the repetition rate of the illumination pulses. Furthermore,
 188 if the system is modified to detect a larger image with more pixels, it would require an MMF with
 189 more modes. In this case, a larger modal dispersion is required that makes the waveforms becoming
 190 broader so that the pulses can be split adequately for separating the information in more modes.
 191 Next, the broadening of the waveforms will cause the reduction of the frame rate. Thus, considering
 192 that the number of modes are in direct proportion to the number of resolvable pixels in the images,
 193 the frame rate will be approximately inversely proportional to the number of resolvable pixels.⁴¹

For megapixel imaging, if we want to detect images containing such a large number of pixels, the frame rate must be reduced to $1/(10^6 \text{ pixel} \times 46.8 \text{ ps/pixel}) = 21.4 \text{ kfps}$, which is similar to the result obtained by the reviewer. Thus, the reviewer's remark on our system is very appropriate: it has no advantage in imaging speed for detecting images large sums of pixels. This conclusion is also clear in **Response 4**, where we discussed that the data rate of our system is similar to that of fast cameras operating at relatively lower frame rates.

However, we emphasize that, as discussed in **Response 4**, the pixel rates of fast cameras are predicted to be lower than that of our approach when operating at frame rates $> 1 \text{ Mfps}$, at which the image size that we can achieve will be $< 21.4 \text{ G} / 1 \text{ Mfps} = 21.4 \text{ k pixels}$. This means that, our system only has higher fame rates than the fast cameras when used to detect images of sizes $< 21.4 \text{ k pixels}$. When the image size exceeds 21.4 k pixels, our approach does not have any advantage in terms of imaging speed.

In fact, our system has great advantages over traditional cameras in many other aspects such as high flexibility, a small volume and a wide operational wavelength band as discussed in the introduction of the manuscript.

Comment 6: The authors mention fluoride fibers as a potential use case, however these fibers have a loss of the order of 0.1 dB/m and therefore, this limits the fiber length to tens of meters which would be too short to produce a temporal spread. Clarifications are here necessary.

Response 6:

We agree with the reviewer that the loss is an important factor to consider when stepping to the mid infrared wavelengths using the fluoride fibers as the modal dispersion medium.

Although today's technology limits the loss of fluoride fibers to 0.1 dB/m, our method still has the potential to be adopted in this situation. In **Response 4** above we validated that for the current system, a slightly reduced but still acceptable imaging performance can be maintained even when the length of the MMF is reduced to 150 m. In this case, the total optical loss will be significantly reduced (compared with the previous 1 km fiber system), and the feasibility of the use of fluoride fibers as a modal-dispersion medium in such systems is drastically enhanced. Thus, we believe that this system has the potential to be used in the mid infrared bands, considering the points below:

- 1) We expect technology improvements to lower the loss of fluoride fibers. According to the calculations of Shibata et al. (Shibata et al., Electronics Letters 17(21), 2007), the intrinsic losses in BaF₂-GdF₃-ZrF₄ fibers and BaF₂-GaF₃-YF₃-AlF₃ fibers are predicted to be 10⁻³ dB/km @3.4 μm and 10⁻² dB/km @2.7 μm, respectively. Therefore, there is still room to reduce fiber loss if the nonintrinsic loss caused by impurity absorption or defect scattering can be further reduced with the development of fiber production technology. This may help lower the loss to an acceptable level for the use of fluoride fibers in our system.
- 2) Some new emerging types of hollow-core photonic crystal fibers also present lower losses in the MIF bands. These nanostructured fibers lower the losses to the order of 0.01 dB/m (see the table below); moreover, the working wavelengths of such nanostructured fibers are configurable, making it easy to shift the central wavelengths to the desired values.

DOI	Types	Loss
10.1109/JLT.2014.2311037	Hollow core Bragg fiber	0.01 dB/m @3μm~4.5μm 10 ⁻⁸ dB/m @3.5μm (simulation)
10.1364/CLEO_SI.2018.SF1K.2	Kagome-lattice structure fiber	0.025 dB/m @2 - 2.5μm
10.1364/OE.20.011153	Negative curvature fibers fiber	0.034 dB/m @3.05μm
10.1364/OPTICA.3.000218	Negative curvature fibers fiber	0.025 dB/m @3μm
10.1063/1.5115328	Anti-resonant fiber	0.018 dB/m @3.1μm

- 3) In terms of our system itself, there are several aspects that can be modified to reduce the fiber length requirement. The first involves using shorter pulses and faster photodetectors, which will promote the temporal spread of the pulses in the MMF. Second, we can increase the intermodal dispersion by using MMFs with higher NAs. Considering the average delay between neighboring modes (Gloge D., Applied optics, 1971, 10(10)):

$$\Delta t_{av} = \frac{\Delta \tau_{gr}}{N} = \left(1 - \frac{2}{v}\right) \frac{\Delta n L}{cN}$$

- 4) Here, $\Delta \tau_{gr}$ is the total group delay of all modes, $v = akNA$ is the normalized frequency, N is the mode number, L is the fiber length, and Δn is the refractive index difference between

the core and cladding. For an MMF with a large v , we have $N = v^2 / 2$. Thus, if the mode number N is fixed, the average delay Δt_{av} will only be determined by Δn , which depends on the NA. In addition, the relationship is linear. Hence, increasing the NA can reduce the fiber length requirement.

- 5) Moreover, in the original manuscript, we discussed that by splicing a multimode fiber amplifier to the end of the MMF, weak signals can be amplified, which can relax the fiber loss requirement.

We have supplemented detailed clarification in the discussion section of the manuscript:

311	CMOS cameras are sensitive only to wavelengths below $1.1 \mu\text{m}^{23}$. Our method also has the potential
312	to operate in the mid-infrared or THz bands, considering the maturity of photodetectors in these
313	bands. In the mid-infrared band, the use of a fluoride-glass fiber can significantly reduce optical
314	loss, which makes it possible to develop long waveguides with high intermodal dispersion in these
315	bands. In addition, we can see from Fig. 4b that the MMF length may be potentially reduced to 150
316	m with little degradation of image quality. The required MMF length can be further reduced by
317	increasing the NA of the MMF, which increases the intermodal dispersion. Thus, although current
318	technology can only fabricate fluoride-glass fibers with losses on the order of 0.1 dB/m, ⁴⁰ the total
319	loss can be controlled to acceptable range. In addition, there is still room to lower the loss in fluoride-
320	glass fibers according to the theoretical predictions of Shibata et al ⁴¹ . Thus, it is possible to extend
321	the wavelength of this system to the mid-infrared region. Also, in the THz band, much work has

Response to Reviewer #2

Comment 1: In the abstract, the authors claim “high mechanical flexibility” of their system. However, it’s well known that a transmission matrix of a multimode fiber is very sensitive to fiber deformations. How robust is the proposed approach to fiber bending or temperature changes or other external factors? How stable is the system? I guess, it’s required to repeat the training procedure if the fiber probe and/or MMF has been moved. Could you discuss it in the manuscript.

Response 1:

We sincerely thank the reviewer for carefully assessing our manuscript and raising this constructive comment. The suggestion was well taken. We have performed supplementary experiments to discuss and validate the robustness of our proposed systems.

We agree with the reviewer that external factors will have some impact on the performance of our system. Nonetheless, our supplementary experiments have verified that we actually do not need to repeat the training process within a certain range of external changes.

To analyze the robustness, we supplemented two experiments to explore the effect of temperature and bending on the system performance as detailed below.

(1) Temperature effect

To change the environmental temperature, we adjusted the air conditioner to different temperatures and used a mercurial thermometer to record the indoor temperature. In this case, all parts of the system, including the fiber probe, MMF (1km length) and laser source, were affected by temperature changes. First, we collected 10000 image/waveform pairs when $T = 25\text{ }^{\circ}\text{C}$ and they were used to calibrate the system. Then, we changed the temperature from $23\text{ }^{\circ}\text{C}$ to $27\text{ }^{\circ}\text{C}$ and collected waveforms of the test images at different temperatures, with 500 data points for every $0.5\text{ }^{\circ}\text{C}$ change. These waveforms were used to test the system. The qualities of the restored images are shown in Fig. R9 by the blue line. We see that the system has high sensitivity to temperature. If we prescribe that recalibration is required when the average fidelity is reduced to less than 70%, the system must be recalibrated when the temperature change exceeds $0.5\text{ }^{\circ}\text{C}$.

However, we verified that this sensitivity can be inhibited by joint training. Joint training uses data collected at different temperatures to co-train the network to adapt to image reconstruction at different temperatures. This method uses the advantages of the neural network, which are inapplicable to conventional transmission-matrix imaging methods. To demonstrate this, in the calibration stage, we first collected 10000 training samples at a fixed temperature of $T = 25\text{ }^{\circ}\text{C}$. Then, we continuously adjusted the temperature from $23\text{ }^{\circ}\text{C}$ to $27\text{ }^{\circ}\text{C}$, and we collected another 4000 training data points at different temperatures. These 14000 pairs of data were used to co-train the network. After training, the network was used to recover another 4000 test images detected at different temperatures. The results are shown in Fig. R9 below by the red line. We found that the quality of the images recovered at different temperatures was largely improved. The fidelity remained above 70% within the range of $23.5\text{ }^{\circ}\text{C}$ – $26.5\text{ }^{\circ}\text{C}$, which means that recalibration is not required within a $3\text{ }^{\circ}\text{C}$ range. In addition, this range is expected to be further expanded if more than 4000 training data points are collected at different temperatures in the calibration stage.

Fig. R9 Imaging performance at different temperatures. The blue line shows the results when the system was calibrated at a fixed temperature. The red line shows the results when joint training was used.

Example images restored at different temperatures using joint training are shown below. We see that the images could be restored fairly well from $T = 23.5\text{ }^{\circ}\text{C}$ to $T = 26.5\text{ }^{\circ}\text{C}$. Considering that the general indoor working temperature is within this range, our system has certain practicability. In practical applications, the 1-km MMF and the laser source can be packed in a thermostatic container, which can further lower the sensitivity to the environmental temperature.

Fig. R10 Example images recovered at different temperatures after joint training.

(2) Bending effect

In real applications, the long MMF can be immobilized because it only acts as a modal-dispersion medium, and there is no need to move it. Thus, we only considered the bending of the fiber probe. In this experiment, the bending state is schematically shown below with the fiber probe bent into a semicircle, and we used the bending radius R to measure the degree of bending.

Fig. R11 Sketch of the bending state.

Similarly, 10000 image/waveform pairs were used to calibrate the system when the bending radius

was fixed at 28 cm. Then, the fiber was bent again, and the radius was changed from 28 cm to 12 cm. We collected test data under 8 different bending states, with 500 data per state. These data were used for testing. The quality of the restored images at different bending states is shown in Fig. R12 by the blue line. We can see that the image quality decreased obviously as the bending radius decreased. When the radius was below 22 cm, the quality deteriorated to below 70%, which means that the system will require recalibration when the bending radius changes by 5 cm. The system's sensitivity to bending is probably caused by fiber-stress-induced modal crosstalk inside the fiber core, which influences the shapes of the waveforms.

Fig. R12 Imaging performance at different bending radii. The blue line shows the results when the system was calibrated at a fixed bending state. The red line shows the results when joint training was used.

This deterioration can also be improved by joint training. We demonstrated this in the same way by changing the bending radius from 28 cm to 12 cm and collecting an additional 4000 training data points at different bending states. The initial 10000 data points and the additional 4000 data points were combined to co-train the network. Then, the system was used to recover other test images detected at different bending radii. The results are shown as the red line in Fig. R12. We see that the bending effect was greatly weakened. The fiber probe can bend from $R = 28$ cm to 19 cm without the need for recalibration. Example images restored at different bending states are shown in Fig. R13. We see that the images could be mostly restored until $R = 19$ cm.

Fig. R13 Several recovered images at different bending radius after joint training.

We have added a new section in the manuscript to discuss the robustness in the manuscript:

284 **System robustness.** To analyze the robustness of the system, we investigated the influence of
 285 temperature and fiber bending on the imaging performance. For temperature effect, we changed the
 286 environmental temperature from 23 °C to 27 °C by adjusting the air conditioners. We tested the
 287 imaging performance at different temperatures in two different cases. In the first case, the neural
 288 network was trained with image/waveform pairs collected with the temperature fixed at
 289 approximately 25 °C. In the other case, the network was trained with the data collected at different
 290 temperatures, called joint training. The details of the experiment and test results are shown in
 291 Supplementary Note 6. While the average fidelity of the recovered images remains above 70%
 292 within a temperature variation of 0.5 °C in the first case, this variation range increases to 3 °C for
 293 the joint training case. This temperature sensibility is mainly caused by the temperature-induced
 294 index-distribution change in the MMF which influence the temporal distribution of the subpulses.
 295 To investigate the bending effect, we fixed the fiber-end ball and bent the fiber probe into a
 296 semicircle with variable radii as shown in the inset of Fig. S11b. Similarly, we tested the imaging
 297 performance under different bending states R in two cases: training with the data collected under
 298 one bending state ($R = 28$ cm) and joint training with the data collected under different bending
 299 states. The results in Fig.S11 shows that in the first case, the average fidelity can remain above 70%
 300 when the bending radius changes from 28 cm to 22 cm. And for the joints training, the average
 301 fidelity can remain above 70% as the fiber probe is bent from $R = 28$ cm to 19 cm. The sensibility
 302 to the bending is mainly caused by the fiber-stress-induced modal crosstalk inside the MMF. In
 303 summary, we verified that this system has certain robustness for practical applications.⁶⁴

And the experimental details are added to the Supplementary Note 6 of the supplementary materials.

Comment 2: Could the authors provide more information on the image recovery procedure. How long is the training step? How many waveform/image pairs are needed for different datasets? How much time does it take to recalibrate the system for a new object type (both measurements and training)?

Response 2:

Thank you very much for the constructive suggestions. The details of the calibration procedure are as follows.

The procedure includes 4 steps as shown below.

Fig. R14 Calibration and testing procedure and the time required for every step.

The first step is collecting waveform/image pairs for training. To explore the minimum number of pairs required for recovery, we trained the network with different numbers of samples, and the results are shown in Fig. R15. We found that using 10000 samples is adequate to achieve the optimal performance. Thus, at least 10000 waveform/image pairs should be collected. We set the oscilloscope to automatically collect waveforms of different images at a speed of 0.25 sec/image. The whole collection procedure required **42 minutes**.

Fig. R15 Performance when the network was trained with different samples.

Then, the collected waveform data must be processed before feeding into the network. Raw waveform data may include several periods. We selected one period and converted the waveforms of all images into a MATLAB matrix. This matrix was processed by a MATLAB program, which required **5 minutes**.

Next, the waveform/image samples are fed to the U-Net network for training. The training process with 10000 samples is shown in Fig. R16. The model converged after approximately 15 iterations, which required approximately **4 minutes**. We used the online computing resource from Google Colab (<https://colab.research.google.com/>) with a Tesla P100-PCIE GPU for training and testing.

Fig. R16 Training process.

Finally, the trained model is ready to transform newly detected waveforms into images. In conclusion, the whole calibration process required **51 minutes**.

The collection procedure takes up the most time, which is caused by the difficulty of synchronizing the fast operations of image switching in the DMD and data saving in the oscilloscope. We believe that the collection time can be greatly decreased if specialized software is developed to control the DMD and oscilloscope simultaneously at a high speed.

In addition, we have verified in the following responses that our approach has certain generalization ability, and through further optimization, it will be able to image different kinds of objects without recalibration (see the details in **Responses 4-6**).

We have supplemented the description of calibration details in the manuscript:

369 During the process of collecting waveforms of the training images, we set the oscilloscope to
370 automatically save waveforms at a speed of 4 waveforms per second. We verified that using 10000
371 samples was adequate to achieve the optimal performance (see Supplementary Note 8). Thus, the
372 whole collection spanned 42 minutes. The collected raw data was processed before feeding to the
373 neural network. Because the time signal included several periods of waveforms, only one period
374 should be selected and finally the waveform data of all images were converted into a MATLAB
375 matrix. This was processed by a MATLAB program, requiring approximately 5 minutes. Training
376 the U-Net network with 10000 sample data required approximately 4 minutes. We used the online
377 computing resource from Google Colab (<https://colab.research.google.com/>) that provides a Tesla
378 P100-PCIE GPU. In conclusion, the whole calibration process, including sample collection,
379 processing and training, required approximately 51 minutes.⁴⁹

And more detailed processes are supplemented to the Supplementary Note 8 of the supplementary materials.

Comment 3: Could the authors provide more information on the experimental setup. What model of a DMD has been used? What photodetector (photodiode) and oscilloscope have been used? Key parameters of all the elements should be provided.

Response 3:

Thanks for the important suggestions.

We have added all the key parameters of the elements that we used, including those listed below:

DMD: Texas Instruments DLP4500. The surface consists of 912×1140 micromirrors, each with a size of 7.56 μm.

Photodetector: Thorlabs DXM30BF. The photodetector has a 30 GHz response bandwidth and a 15 ps impulse response. The sensible spectrum is 750 nm–1650 nm. It has a OM4 (50/125 μm) fiber coupling input.

Oscilloscope: Tektronix MSO73304DX. The oscilloscope has a 33 GHz analog bandwidth and an analog sample rate of 100 G/s.

We have added the model information of these instruments to the Methods section of the manuscript. The maximum record length is 500 Mega samples.

Fiber probe: NUFERN FUD-4658, BD-S50/70/360-22FA-HP triple-cladding fiber. The diameters of its core, first cladding and second cladding are 50 μm, 70 μm and 360 μm, respectively. The core is a step-index type and has an NA of 0.2.

MMF: 50/125 μm step-index fiber with an NA = 0.22.

Laser source: Homemade mode-locking fiber laser. The laser operates at a wavelength of 1064 nm with a 3 dB bandwidth of 0.11 nm. The full width at half maximum of the output pulse is 26.4 ps. The average output power is 1 W.

We have added these key information to the main text:

105 **Experimental setup.** The structure of the system is illustrated in Fig. 2a. The illumination pulses
106 from a mode-locked fiber laser are directly coupled into the fiber probe by a side-pump coupler³¹.
107 After approximately 2 meters of transmission, the illumination pulses emerge from the fiber probe
108 to illuminate the intensity patterns displayed by a digital micromirror device (DMD). The pulse laser
109 operates at a wavelength of 1064 nm with a 3 dB bandwidth of 0.14 nm. While the full width at half
110 maximum of the output pulse is 26.4 ps, it broadens to 45.1 ps after the pulses transmit through the
111 fiber probe and emerge from the fiber-end-ball (see Supplementary Note 1). Then, the light reflected
112 from the patterns reenters the fiber probe, as shown in Fig. 2c. In this way, the illumination and
113 reception of light are integrated into a single fiber probe. The other end of the fiber probe is spliced
114 with a 1-km MMF (50/125 μm , numerical aperture (NA) = 0.22), in which the spatial information
115 carried in the signal pulses is transformed into temporal waveforms. The step-index core of the
116 MMF can provide much greater intermodal dispersion than a graded-index core³². In such a long
117 MMF, the delay differences between different modes are sufficiently large to cause each signal pulse
118 to split into a burst of subpulses. The temporal waveforms of the pulses at the other end of the MMF
119 are detected by an ultrafast InGaAs photodetector (spectral response 750–1650 nm, bandwidth 30
120 GHz) and instantly stored in the memory of an oscilloscope (100 G samples/sec.). In the training
121 stage, different displayed images and the corresponding waveforms are used to train the neural
122 network model. After training, the network is capable of recovering new images directly from the
123 acquired waveforms, as shown in Fig. 2b.^{4†}

353 **Methods^{4†}**

354 **Experiments.** The laser source is a homemade Yb-doped mode-locked fiber laser with a spectral
355 width of 0.2 nm. The average output power is 1 W. The fiber probe is a triple-cladding fiber with
356 dimensions of 50/70/360 μm (NUFERN FUD-4658, BD-S50/70/360-22FA-HP). The NAs of the
357 step-index core and the second cladding layer of the triple-cladding fiber are 0.2 and 0.46,
358 respectively. The homemade fiber coupler couples light from the source into the second cladding
359 layer of the probe. The fiber-end ball at the end of the probe was produced via fusion with a fusion
360 splicer. Because we adopted a pulse laser, considering that the DMD can perform only grayscale
361 modulation of continuous light, all images were binarized before loading into the DMD. The DMD
362 (Texas Instruments DLP4500) consists of 912×1140 micromirrors, each being 7.56 μm in size. The
363 photodetector (Thorlabs DXM30BF) has a 30 GHz response bandwidth and a 15 ps impulse
364 response. The sensible spectrum is 750–1650 nm. The photodetector receives light through an OM4
365 (50/125 μm) fiber, which is connected to the other end of the MMF. The oscilloscope (Tektronix
366 MSO73304DX) has a 33 GHz analog bandwidth and a sample rate of 100 G/s. The maximum record
367 length is 500 Mega samples.^{4†}

Comment 4: As far as I can see, the proposed approach allows to reconstruct the images if it's known what kind of object we are looking at. I would call it content-aware kind of imaging. I think this limitation should be highlighted [in the abstract] and discussed.

Response 4:

We thank the reviewer for raining this constructive comment. This suggestion was well taken.

According to our experiments, we found that the current method introduced in the manuscript does have such a limitation. However, we further proved that the generalization of the neural network used to reconstruct the images can be largely improved to realize imaging of content -unknown objects.

The low generalization of the original system can be improved in the following two aspects:

- 1) Although the U-Net model can realize high-quality image reconstruction for objects of the same type, we found that its generalization is not as high as that of a fully connected network.
- 2) The generalization of a neural network can be much improved if it is fed richer samples. This requires adopting various kinds of images to train the model instead of using only one kind of image from one dataset. Our experiments show that a high generalization can be achieved by using images containing almost random patterns.

Thus, we used a group of images from the Omniglot dataset, in which the images are made up of nearly random lines, to calibrate the system. Some of these images are shown below.

Fig. R17 Example images in the Omniglot dataset.

The collected waveforms of 20000 images of this kind were used to train the network. Then, the trained model was used to restore images of objects of completely different types. The results are shown in Fig. R18. Although the test images were of completely different types, they could still be partially recovered, which indicates that this system has certain generalization ability.

Fig. R18 Exampled recovered images of different types.

Although the image qualities may not be as high as those of the content-aware images, there is still potential to further improve the imaging quality of the current system by adopting MMFs with more modes or implementing various measures to reduce noise in the signal waveforms.

We have added the statement of this limitation in the revised abstract, along with the description of our partial capacity of imaging random objects:

13 opposite end of the fiber. The demonstrated scheme could not only recover images that are of the
 14 same category as the training set with high quality, but also could recognize images that are of
 15 different categories as the training set with slightly reduced quality. The fiber probe could directly

Also, the discussion about the partial ability of imaging objects with different kinds are added to the manuscript:

200 In the imaging experiments discussed above, the detected images are of the same type as the images
 201 used to train the network. However, we verified that this system could also recover images of
 202 different types. To validate this, we replaced the U-Net network with a fully connected network. We
 203 found that for the current system, although the U-Net model could realize high-quality imaging of
 204 type-aware objects, its ability to image random types of objects, i.e., generalization ability, was not
 205 as high as that of the fully connected network (see Supplementary Note 4 for details). Moreover,
 206 given that the generalization of a trained neural network is closely connected to the complexity of
 207 the training data, we used the images from the Omniglot dataset
 208 (<https://github.com/brendenlake/omniglot/>) for training. These images contain patterns made up of
 209 nearly random lines. In addition to the original images from the Omniglot dataset, we also generated
 210 new training images based on the original images by shifting, rotating or overstriking the original
 211 patterns to increase the complexity of the training data. Finally, 20,000 images with different
 212 complex patterns were used for the training. Then, the trained model was used to restore images that
 213 were completely different from the training images. Exemplary results are shown in Fig. 5a and the
 214 test images could be recovered with high fidelities. Furthermore, we also verified that this system
 215 can be used for detecting grayscale images (see Supplementary Note 9 for details).

a											
True											
Recovered											
SSIM:		0.52	0.63	0.53	0.48	0.75	0.50	0.56	0.54	0.61	0.55
Fidelity:		84.3%	81.6%	67.0%	81.9%	86.1%	74.2%	84.9%	80.5%	80.8%	83.4%

And some experimental details are also added to the Supplementary Note 4 of the supplementary materials.

Comment 5: The authors demonstrate only binary (black-and-white) images. Does the proposed approach allow to image more complex gray scale objects?

Response 5:

We thank the reviewer for raising this important comment. We have performed supplementary experiments to demonstrate that our proposed systems can also further modified to distinguish more complex gray scale objects. The details are as the following.

Obviously, the waveforms contain some gray scale information of the objects, because the gray scale has a direct correspondence to the intensity of the time signals. The reason why we did not use grayscale images is that the micromirrors of the DMD rely on rapid vibration to produce grayscale modulation, which is only applicable for CW lasers. Our laboratory lacks reflective gray-modulation SLMs. Therefore, it is difficult for us to calibrate the system with grayscale images. However, there is an alternative solution that can display pseudo grayscale images on the DMD. Because a pixel in the DMD can only present 1 or 0, to display a pseudo grayscale image, we transformed a grayscale image into pixel blocks. In detail, we used a 3×3-pixel block on the DMD to represent one pixel of a grayscale image. According to the number of pixels with a value of “1” in the 3×3 block, the pixel block can present 9 levels of grayscale (0-9), as shown in Fig. R19. Because the resolution of the displayed images (192×192) is far beyond the resolution of our system (approximately 20×20, as discussed in **Response 6**), a 3×3 pixel block can act as one grayscale pixel, where the reflected

intensity of light is direct proportional to the number of pixels with a value of “1” in this block.

[REDACTED]

Fig. R19 Explaining pseudo grayscale images.

Because the pseudo grayscale images (192×192) require a much larger region on the DMD and the single fiber end can only detect a small region on the DMD, we used a lens in front of the fiber end to expand the illumination beam, as shown in the figure below. Thus, a large region of the DMD surface could be illuminated and detected, and an image containing many more pixels could be displayed.

Fig. R20 Function of an external lens.

In this way, we calibrated and tested the system with the grayscale images from the ImageNet database, which consists of images of different natural scenes. A total of 29000 images were used for training, and another 1000 were used for testing. According to the discussion in **Response 4**, we adopted a fully connected network model instead of the original U-Net model because there were almost no connections between the different images consisting of random natural scenes; thus, a higher-generalization network model was required. Meanwhile, the high complexity of these images helped promote the generalization of the trained model. Example test results are shown in **Error! Reference source not found.**, which proves that grayscale information could still be partially reconstructed. We believe the use of fibers with more modes can help to reconstruct images with more details.

[REDACTED]

Because that we only verified the ability of recovering grayscale information using the pseudo grayscale images, we added these discussion to the Supplementary Note 9 of the supplementary materials instead of in the main text.

Comment 6: Please discuss the spatial resolution achieved. What was the average feature size of the reconstructed images?

Response 6:

Thank you for the constructive suggestion, which was well taken.

Our additional experiments have shown that the smallest spatial resolution of the current system is approximately $15\ \mu\text{m}$ for a $300\times 300\ \mu\text{m}^2$ detection region. The total resolvable number of pixels is approximately 20×20 . Limited by the pixel size ($7.56\ \mu\text{m}$) of the DMD, we could not verify the resolution for a smaller field of view. However, because the total resolvable number of pixels for an MMF is fixed, we believe that the spatial resolution will be higher if moving the fiber end closer to the object to decrease the detection region area. Relevant experiments were conducted as described below.

We tested the spatial resolution of our system by detecting images similar to the USAF 1951 resolution target. We used the same network model trained with the 20000 images from the Omniglot dataset to recover the target images. In the original experiment shown in the manuscript, only 28×28 pixels on the DMD surface could be detected by the fiber probe. This region is too small to explore the resolution limit of the system. Thus, we adjusted the location of the fiber-end ball relative to the DMD until it could receive light from a larger region. The area of this region measured approximately $300\times 300\ \mu\text{m}^2$ and included 40×40 pixels. The test images were displayed in this region.

The recovery of some resolution targets is shown in Fig. R21, with the pitches of the bars indicated at the top. We can see from the images in the rightmost column that the smallest bar pitch that could be distinguished was approximately $15\ \mu\text{m}$, which was a two-pixel interval. Considering that the image consisted of 40×40 pixels, the total pixel resolution of this system is approximately 20×20 . In the last row of images, we show the recovery results for dots of different sizes. The smallest dot had a diameter of a two-pixel interval, indicating that the smallest feature that can be recognized by the system is approximately $15\ \mu\text{m}$.

Although the current resolution is not very high, it has the potential to be improved by using an MMF/fiber probe with a higher NA or a larger core area with more modes. Principally, the fiber modes will be quadrupled if the NA or core diameter is doubled. Considering its ultrahigh imaging speed, our method still has high practical value.

Fig. R21 Recovered images of the resolution targets and small dots.

This discussion is also supplemented to the manuscript:

216

217 **Fig. 5 (a)** Several recovered images of different types. **(b)** Recovered results of images containing resolution targets

218 similar to the USAF 1951. **(c)** Vertical intensity profile of the smallest recovered bars along the red dotted line.⁴⁴

219 ⁴⁴

220 Next, we used the same trained model to test the spatial resolution of the system by recovering

221 images of resolution targets similar to the USAF 1951. Here, we adjusted the location of the fiber-

222 end-ball relative to the DMD until it could receive light from a larger region of the DMD surface.

223 The area of this region measured approximately $300 \times 300 \mu\text{m}^2$ and included 40×40 pixels (pixel size

224 of $7.56 \mu\text{m}$), which was larger than the previous 28×28 pixels. This could help us to explore the

225 resolution limit of the system. The images of resolution targets contain white bars with different

226 pitches. The recovered results are shown in Fig. 5b, indicating that the smallest bars with pitches of

227 $15 \mu\text{m}$ (occupying two pixels on the DMD surface) could be distinguished, which is shown more

228 clear in Fig. 5c.⁴⁴

REVIEWER COMMENTS

Reviewer #1 (Remarks to the Author):

The authors conducted convincing additional measurements of the fidelity of the image reconstruction at different fiber length and thoroughly discussed the imaging speed as a result. the manuscript is now much more complete and to my satisfaction.

Reviewer #2 (Remarks to the Author):

The authors have addressed all the questions raised in the previous reviews.

The description of the experimental approach is significantly improved in the revised manuscript and the SI. Overall, I think this is a commendable advancement and therefore I recommend publication.